# Genetic Pathogenesis of Inflammation-Associated Cancers in Digestive Organs

**DOI:** 10.3390/pathogens10040453

**Published:** 2021-04-09

**Authors:** Risa Nakanishi, Takahiro Shimizu, Ken Kumagai, Atsushi Takai, Hiroyuki Marusawa

**Affiliations:** 1Department of Gastroenterology, Red Cross Osaka Hospital, Osaka 543-8555, Japan; r_nakanishi@kuhp.kyoto-u.ac.jp; 2Department of Gastroenterology and Hepatology, Graduate School of Medicine, Kyoto University, Kyoto 606-8501, Japan; shimy@kuhp.kyoto-u.ac.jp (T.S.); kenk833@kuhp.kyoto-u.ac.jp (K.K.); atsushit@kuhp.kyoto-u.ac.jp (A.T.)

**Keywords:** mutation, IBD, colitic cancer, GERD

## Abstract

Epidemiological, clinical, and biological studies convincingly demonstrate that chronic inflammation predisposes to the development of human cancers. In digestive organs, inflammation-associated cancers include colitis-associated colorectal cancers, *Helicobacter pylori*-associated gastric cancer, as well as Barrett’s esophagus and esophageal adenocarcinoma associated with chronic duodenogastric-esophageal reflux. Cancer is a genomic disease, and stepwise accumulation of genetic and epigenetic alterations of tumor-related genes leads to the development of tumor cells. Recent genome analyses show that genetic alterations, which are evoked by inflammation, are latently accumulated in inflamed epithelial cells of digestive organs. Production of reactive oxygen and aberrant expression of activation-induced cytidine deaminase, a nucleotide-editing enzyme, could be induced in inflamed gastrointestinal epithelial cells and play a role as a genomic modulator of inflammation-associated carcinogenesis. Understanding the molecular linkage between inflammation and genetic alterations will open up a new field of tumor biology and provide a novel strategy for the prevention of inflammation-associated tumorigenesis.

## 1. Introduction

Cancer is induced by factors such as chemical carcinogens, infection, and chronic inflammation. Numerous epidemiological studies show that chronic inflammation predisposes to tumor formation, including colorectal cancer caused by inflammatory bowel disease (IBD), gastric cancer caused by *Helicobacter pylori* (*H. pylori*) infection, and esophageal cancer associated with gastro-esophageal reflux disease (GERD). Accordingly, >25% of human cancers may be associated with inflammatory conditions [1]. Chronic inflammation could contribute to the promotion and progression of cancer by various mechanisms. It is important to note that tumor cells arise from the stepwise accumulation of genetic and epigenetic alterations of diverse genes under conditions of constitutive inflammation (Figure 1). The molecular mechanisms of the acquisition of genetic alterations under inflammatory conditions are unknown, although recent advances in genome technology, such as ultra-deep sequence analyses, have increased our understanding of the genetic and epigenetic aberrations accumulated during the development of inflammation-associated cancer. This review highlights the genetic pathogenesis of inflammation-associated human cancers of digestive organs and summarizes our current knowledge.

## 2. Colitis-Associated Colorectal Cancer

### 2.1. Clinical Feature and Risk Factors

Colon and rectal cancer are the fourth and the seventh most common cancer worldwide, respectively [2]. Colorectal cancer comprises major carcinogenic pathways, including the adenoma-carcinoma sequence, the serrated pathway, and colitis-associated colorectal cancers (CACs) of patients with IBD. IBD is defined as chronic intestinal inflammation associated with autoimmunity and gut bacteria and typically refers to ulcerative colitis (UC) and Crohn’s disease. A cohort study of patients with IBD revealed that chronic inflammation of the large intestine increases the cumulative risk of CACs. CACs that develop in patients with IBD exhibit distinct features compared with those with sporadic colorectal cancers. For example, CACs may be derived from dysplasias, while the majority of sporadic colorectal cancers arise from adenomatous polyps. Indeed, the risk of subsequent colorectal cancer is relatively low after endoscopic removal of a polypoid dysplasia of patients with UC.

The incidence of CACs is associated with the extension, duration, and severity of colonic inflammation [3]. UC patients with left-sided colitis have an intermediate risk, and those with pancolitis have a maximal risk for the development of CACs, while patients with ulcerative proctitis are not at increased risk [4]. In addition, UC patients with primary sclerosing cholangitis also have a risk of CACs [5]. The overall incidences of CACs of patients with UC are approximately 0.2%, 0.7%, and 1.2% after 10, 20, and 30 years, respectively [6]. The decade-specific incidences of CACs correspond to cumulative risks of 1.6%, 8.3%, and 18.4% after 10, 20, and 30 years, respectively [6]. Furthermore, CACs frequently develop at multiple sites in the inflamed colon of patients with IBD [3], and the lengthy duration of UC is frequently associated with the development of precancerous dysplasia of the inflammatory mucosa. Accordingly, European Crohn’s and Colitis Organisation guidelines refer that patients with high risk factors, such as extensive colitis with severe active inflammation, stricture or dysplasia detected within the past five years, primary sclerosing cholangitis, should have their next surveillance colonoscopy scheduled for one year. Patients with intermediate risk factors, such as extensive colitis with mild or moderate active inflammation, should have their next surveillance colonoscopy scheduled for two to three years [7]. These findings suggest that chronic inflammation of the colonic epithelium plays a critical role in tumorigenesis.

### 2.2. Genetic Alterations Accumulated in Inflamed Epithelial Cells

Sporadic colorectal cancer and CACs undergo multistep development driven by the accumulation of genetic aberrations; however, their mechanisms of acquisition of mutations during carcinogenesis differ greatly. Sporadic colorectal cancer develops via the adenoma-carcinoma sequence and tumor initiation caused by *APC* mutations followed by mutations in other genes, such as *KRAS*, *TP53*, *PIK3CA*, and *BRAF* [8]. In contrast, in CACs, precancerous “dysplasia” arises multifocally, driven by initial loss or mutation of *TP53* [9,10,11,12], followed by the acquisition of somatic mutations in tumor-related genes, such as *KRAS* and *APC* [13]. Therefore, the initial *TP53* alteration is an important “trunk” mutation in the development of CACs, and *TP53* mutations are detected in most of the cells located at the crypt in dysplastic tissue [12]. Allelic loss of *TP53* occurs in 50–85% of patients with CACs, and the frequency of the *TP53* alterations correlates with the atypia of the tumor [10,14].

Evidence indicates that reactive oxygen and nitrogen species (RONS) released from inflammatory cells cause the genetic alterations acquired during the development of CACs. The production of RONS and nitric oxide synthase (NOS) increases in the inflamed colonic mucosa of patients with UC; therefore, chronic inflammation of the colon may be closely associated with oxidative stress [15,16]. Somatic mutations are acquired during DNA replication through the reactivities of molecules associated with RONS, such as dinitrogen trioxide, peroxynitrite, and 8-oxo-2’-deoxyguanosine, which may induce genetic alterations [17,18,19].

In contrast, enzymes that possess nucleotide editing activity may be involved in the production of genetic alterations that lead to tumorigenesis. Such enzymes include members of the apolipoprotein B mRNA-editing enzyme, catalytic polypeptide-like (APOBEC) family. Evidence indicates that APOBEC-family molecules play important roles in maintaining homeostasis and the immune response by inducing somatic mutations in targeted DNA or RNA sequences [20]. Among them, activation-induced cytidine deaminase (AID) induces genetic changes in human DNA sequences. AID is specifically expressed in activated B cells under physiological conditions and contributes to somatic hypermutation and class switch recombination in antigen-driven immunoglobulin genes [21,22]. Aberrant expression of AID in inflamed colonic mucosa contributes to the development of CACs [23,24]. CAC tissues, as well as inflammatory noncancerous colonic mucosa, highly express AID via the TNF-α-nuclear factor (NF)-κB pathway or STAT6-dependent pathway activated by a Th2 cytokine, such as IL-4 and IL-13 [23]. Furthermore, IL-10-deficient mice, which serve as a representative model of CACs, develop intestinal tumors at a significantly lower frequency when the AID locus is depleted [24]. These findings suggest that aberrantly expressed AID via chronic inflammation is required to induce genetic alterations during the progression of inflammation-associated colon carcinogenesis.

Recent advances in genome sequencing technology make possible the comprehensive analysis of the colonic mucosa in single crypts. Interestingly, whole exome sequence analysis of single crypts from the colon of patients with UC found that the inflamed colonic mucosa undergoes widespread remodeling through pervasive clones under positive selection by acquiring somatic mutations [25].

Together, these findings suggest that a chronic inflammatory response contributes to the development of CACs through the accumulation of genetic alterations in the inflamed colonic epithelium, possibly through the production of RONS, upregulation of nucleotide-editing enzymes, or both.

## 3. Gastric Cancer Associated with *Helicobacter pylori* Infection

### 3.1. Clinical Feature and Risk Factors

Gastric cancer is the fifth most common cancer and the second most common cause of cancer-related death worldwide [2]. Causative factors associated with the development of gastric cancer include diet (salt and salt-preserved food), smoking, obesity, GERD, Epstein-Barr virus (EBV), and *H. pylori* infections. Epidemiological studies convincingly demonstrate that the strongest risk factor for the development of gastric cancer is *H. pylori* infection and the resultant chronic gastric inflammation [26]. In 1994, the World Health Organization/the International Agency for Research on Cancer (WHO/IARC) classified *H. pylori* as a Group 1 carcinogen [27]. In particular, *cagA*-positive strains are considered to be more potent in gastric cancer development. Briefly, CagA is translocated into the cytoplasm of gastric epithelial cells by the type IV secretion system and interacts with and activates pro-oncogenic phosphatase SHP2, followed by the interaction with a variety of human proteins to lower the threshold for neoplastic transformation. The CagA-SHP2 interaction requires phosphorylation of CagA at the Glu-Pro-Ile-Tyr-Ala (EPIYA) motif. The majority of *H. pylori* CagA isolated in East Asia possesses EPIYA-D motif, whereas *H. pylori* CagA isolated in the rest of the world have EPIYA-C motif. Importantly, the interaction of tyrosine-phosphorylated EPIYA-D with SHP2 is stronger than tyrosine-phosphorylated EPIYA-C, contributing to the induction of malignant transformation [28]. Moreover, *H. pylori* can also cause gastric mucosa-associated lymphoid tissue lymphoma in the stomach and several extra-digestive disorders, including immune thrombocytopenia.

The prevalence of *H. pylori* infection, which usually occurs during childhood, is high in countries in East Asia, particularly Korea and Japan. *H. pylori* infection causes chronic gastric inflammation leading to atrophic changes of the gastric epithelium, followed by the development of intestinal metaplasia and intestinal-type gastric cancer, particularly noncardia cancer [29]. Consistently, the incidence of gastric cancer in patients with *H. pylori* infection is 5.9 times higher than that of uninfected patients [29]. Furthermore, eradication of *H. pylori* reduces the occurrence of gastric cancer. For example, the incidence of gastric cancer of asymptomatic patients who received *H. pylori* eradication is lower compared with that of those who did not receive eradication treatment [30]. Of note, *H. pylori* eradication reduces the incidence of subsequent gastric cancer in patients who undergo endoscopic resection of early gastric cancer [30,31]. Moreover, similar to CACs, multicentric tumor development occurs in the *H. pylori*-infected stomach. For example, in patients with early gastric cancer with *H. pylori* infection, 7.5% had multiple synchronous tumors, and 16.9% developed a second metachronous gastric cancer [32]. These findings suggest that the chronically inflamed gastric epithelium caused by *H. pylori* infection possesses sufficient potential for contributing to carcinogenesis.

### 3.2. Genetic Alterations Accumulated in Inflamed Epithelial Cells

Gastric cancer is histologically classified according to Lauren’s classification as intestinal-type and diffuse-type [33]. The former comprises cohesive groups of tumor cells with a glandular architecture and typically develops from gastric mucosa inflamed by chronic *H. pylori* infection [34]. In particular, evidence indicates that intestinal metaplasia is a precursor of intestinal-type gastric cancer. For example, patients with intestinal metaplasia have an annual risk of gastric cancer of 0.1–0.25% [35]. On the other hand, spasmolytic polypeptide-expressing metaplasia (SPEM) has also been highlighted as another metaplastic lesion in the stomach. Studies of animal models show that SPEM generates through the trans-differentiation of chief cells subsequent to the loss of parietal cells [36]. In Mongolian gerbil models, chronic inflammation caused by *H. pylori* infection induces the development of SPEM, which gives rise to intestinal metaplasia and progresses to a further aberrant and invasive phenotype [37]. These data suggest that SPEM represents preneoplastic metaplasia that predisposes gastric cancer, although further studies are required.

Recently, The Cancer Genome Atlas research network demonstrated that gastric cancer can be classified into the subtypes as follows: EBV-positive, microsatellite instability (MSI), chromosomal instability (CIN), and genomically stable (GS) [38]. EBV-positive cancer harbor *PIK3CA* mutations, extreme DNA hypermethylation, and amplification of *JAK2*, *PD-L1*, and *PD-L2*. MSI tumors typically exhibit epigenetic silencing of the mismatch-repair gene *MLH1* and hypermutation. CIN tumors mostly comprise the histologically intestinal type and typically harbor *TP53* mutations, marked aneuploidy, and focal amplification. GS tumors predominantly represent the diffuse histological subtype, and most tumors harbor mutations or fusion of *CDH1* or RHO family genes.

Notably, such genetic alterations occur at the early stage of *H. pylori*-related gastric tumorigenesis. For example, *TP53* mutations and chromosomal alterations frequently occur in patients with early-stage intestinal-type gastric cancer [39,40]. Certain intramucosal gastric carcinomas that develop in the *H. pylori*-infected gastric mucosa exhibit the MSI phenotype with *MLH1* promoter methylation [39]. Importantly, mutations in tumor-associated genes, including *TP53*, are latently accumulated in the inflamed gastric mucosa with intestinal metaplasia [41,42]. These findings indicate that the gastric mucosa exposed to *H. pylori*-associated chronic inflammation acquires genetic alterations that contribute to tumorigenesis.

The mutation signatures of gastric cancer genomes help us understand how somatic mutations are generated during inflammation-associated tumor development [43]. C:G > T:A transitions are most frequently detected in gastric cancer genomes, followed by the C:G > A:T transversion that is associated with reactive oxygen species (ROS) or smoking [38]. C:G > T:A transition mutations typically indicate the involvement of deamination. Spontaneous deamination is the most frequent cause of spontaneously generated mutations and is associated with aging and a lengthened cell cycle. Alternatively, deamination may be induced by the activation of endogenous deaminase AID, which is aberrantly expressed in a substantial proportion of the *H. pylori*-infected gastric epithelium and gastric cancer tissues [44]. In particular, mononuclear cell infiltration and intestinal metaplasia correlate with AID expression [45]. Aberrant AID expression in gastric epithelial cells in vitro induces the accumulation of somatic mutations and chromosomal aberrations [44,46]. Together, lengthier cell cycles, aberrant AID expression, and ROS production in the gastric mucosa caused by chronic inflammation associated with *H. pylori* infection may be required for the accumulation of genetic alterations during gastric carcinogenesis.

## 4. Esophageal Cancer Associated with GERD

### 4.1. Clinical Feature and Risk Factors

Esophageal cancer is the eighth most common cancer and the sixth most common cause of cancer-related death worldwide [2]. Esophageal cancer comprises squamous cell carcinoma and adenocarcinoma. The former predominates, representing approximately 90% of esophageal cancers in the most prevalent geographical areas, such as Iran and North-Central China [47,48]. The frequency of adenocarcinoma of the esophagus has increased, particularly in North America and Western Europe [49,50].

Barrett’s esophagus is a metaplastic change to a columnar-lined epithelium with intestinal-type differentiation from the normal stratified, squamous epithelium of the lower esophagus. Increased exposure of the esophagus to refluxed gastric and duodenal contents observed in patients with GERD may cause chronic esophagitis, and the resultant chronic mucosal damage induces the innate stratified squamous epithelium to transform into the columnar epithelium. Furthermore, Barrett’s esophagus is associated with a high risk of esophageal adenocarcinoma. For example, approximately 10% of patients with GERD develop Barrett’s esophagus, which is a premalignant lesion; and 0.5% of patients with Barrett’s esophagus subsequently experience adenocarcinoma [51,52].

### 4.2. Genetic Alterations Accumulated in Inflamed Epithelial Cells

The transcription factor NF-κB contributes to esophageal carcinogenesis. Gastric acid and bile acid reflux activate NF-κB in a concentration- and time-dependent manner in the nucleus of esophageal epithelial cells [53]. NF-κB mediates inflammation and enhances the expression of genes encoding cytokines, chemokines, growth factors, and apoptotic proteins [54,55]. Moreover, NF-κB contributes to the pathogenesis of Barrett’s esophagus and the development of Barrett’s esophageal cancer [56]. Interestingly, the transcriptional upregulation of AID induced in response to stimulation esophageal epithelial cells exposed to bile acid is mediated by the NF-κB signaling pathway, suggesting that aberrant AID expression represents the link between duodenogastric-esophageal reflux and increased susceptibility to carcinogenesis in the pathogenesis of Barrett’s esophagus [53,57]. Another important factor required for the development of Barrett’s adenocarcinoma includes nitrites, which are converted into nitrous acid and nitrosating species, such as nitric oxide (NO), that occur in the stomach or the esophagus during acid reflux. NO infiltrates epithelial cells, particularly under inflammatory conditions, induces mutation through cytosine-deamination [58], and induces double-strand DNA breaks in Barrett’s esophagus [59].

Genetic analysis of Barrett’s esophagus with dysplasia shows the accumulation of somatic mutations to the same extent as esophageal cancers, although Barrett’s esophagus without dysplasia harbors fewer mutations [60]. *TP53* mutations are frequently detectable in Barrett’s esophagus with high-grade dysplasia, although rarely in nondysplastic Barrett’s esophagus [61]. Furthermore, Barrett’s esophagus harbors functionally significant mutations in tumor-associated genes, such as *TP53*, *DCC*, *CDKN2A*, *SYNE1*, *PRDM9*, *ATM*, *KIF2B*, and *PSMD11* [62], which are frequently mutated in Barrett’s adenocarcinoma [63]. Surprisingly, copy number abnormalities are present in nondysplastic Barrett’s esophagus of some cases with Barrett’s esophagus [64]. For example, focal amplification of the genes encoding the protein tyrosine receptor kinases *ERBB2* and *MET,* as well as those that regulate the cell cycle (*CCND1*, *CCNE1*, and *CDK6*), are detectable in Barrett’s esophagus [62]. The mean number of focal deletions per sample shows a stepwise accumulation from nondysplastic Barrett’s esophagus through dysplastic Barrett’s esophagus to adenocarcinoma [60].

Together, these findings indicate that esophageal epithelial cells exposed to long-term inflammation accumulate genetic alterations, providing a genetic basis for tumorigenesis.

## 5. Conclusions

Accumulating evidence suggests that chronic inflammation triggers genetic alterations and provokes cancer cells (Figure 2). Such evidence was acquired by numerous epidemiological studies that convincingly demonstrate that the suppression of chronic inflammation is closely associated with the prevention of the development of inflammation-associated cancer in certain organs other than the gastrointestinal tract. For example, hepatocellular carcinoma usually develops in the liver, accompanied by chronic hepatitis or cirrhosis. The main cause of hepatitis/cirrhosis is chronic infection with a hepatitis virus, including hepatitis C virus. It is noteworthy that the eradication of the hepatitis C virus by interferon or direct-acting antivirals could prevent the progression of liver fibrosis and reduce the incidence of hepatocellular carcinoma [65,66,67]. We conclude, therefore, that suppression of chronic inflammation by therapeutic intervention will contribute to the inhibition of tumor development by preventing the stepwise accumulation of genetic alterations in inflamed epithelial cells. Further studies of the mechanisms of inflammation-mediated accumulation of genetic aberrations will likely provide novel strategies for preventing the development of inflammation-associated cancer.

## Figures and Tables

**Figure 1 pathogens-10-00453-f001:**
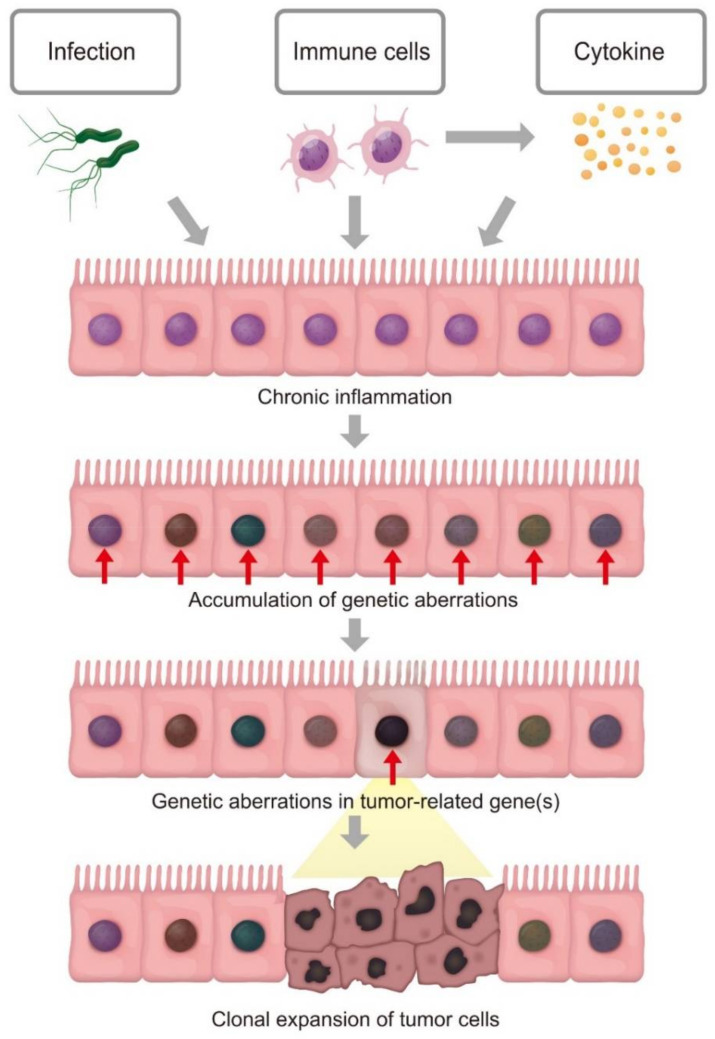
Clonal expansion of inflamed cells with accumulation of genetic aberrations leads to tumorigenesis. Chronic inflammation may cause the occurrence of genetic aberrations in epithelial cells. Each cell independently acquires somatic mutations; however, the inflamed cell that unexpectedly acquires genetic alterations in tumor-associated genes may exhibit a growth advantage and resultant clonal expansion, leading to tumorigenesis.

**Figure 2 pathogens-10-00453-f002:**
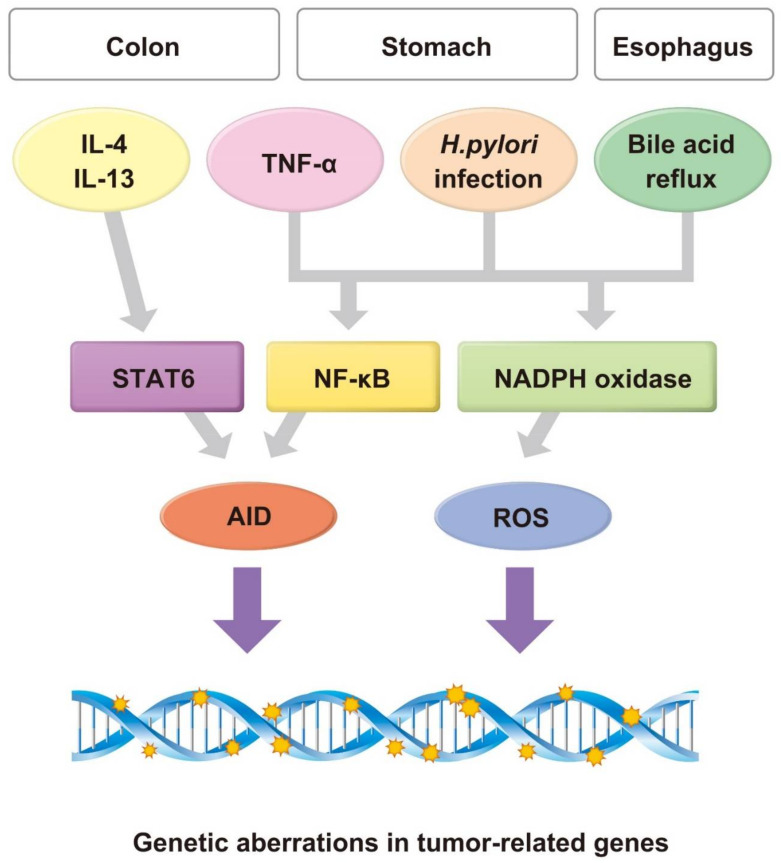
Linkage between inflammation and genetic alterations. Inflammation-related factors, including proinflammatory cytokines (e.g., TNF-α), *H. pylori* infection, and bile acid reflux, trigger activation of the transcription factor NF-κB. NF-κB may induce the activation of the DNA-mutator enzyme, activation-induced cytidine deaminase (AID). Th2 cytokines IL-4 and IL-13 induce the expression of AID via STAT6 activation in colonic epithelial cells. Inflammatory stimulation of each type of cell enhances the production of reactive oxygen species (ROS). AID and ROS may contribute to the generation of genetic alterations, including somatic mutations and copy number aberrations, in epithelial cells.

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
