# Peer review of "Genetic Pathogenesis of Inflammation-Associated Cancers in Digestive Organs"

_pathogens, 2021, doi:10.3390/pathogens10040453_

Round 1

Reviewer 1 Report

In this review, the authors have highlighted the key triggers of human cancers associated with digestive organs. The manuscript is well written and referenced. This is an interesting area of research; therefore, authors should highlight the symptoms, risk factors, and diagnosis in detail. Besides, add a paragraph discussing risk factors. Otherwise, the scope of this review remains limited to few key points.

Abstract

I would suggest revising the abstract, make it more connected.

L12-15: Revise this sentence and change Helicobacter pylori to the italic font.

Line 30, 115, 118, 120, 121, 124, 126,  128, 129, 132, Change thought-out the manuscript “Helicobacter pylori to italic font”.

Colorectal cancer is the fourth most common cancer and the fifth most common cause of cancer-related death worldwide.

L126: Do you mean 5.95? Revise.

L179: in vitro to italic font.

The author contribution section is not updated.

Author Response

We thank the reviewer for his/her thoughtful comments, which have allowed us to improve the paper. We have addressed the reviewer’s criticisms through the following changes to the manuscript.

  1. Authors should highlight the symptoms, risk factors, and diagnosis in detail. Besides, add a paragraph discussing risk factors.

Reply; According to the reviewer’s suggestion, we arranged a paragraph describing “Clinical feature and risk factors” and “Genetic alterations accumulated in inflamed epithelial cells” separately in each cancer.  In addition, we revised description about the risk factors of colitis-associated colorectal cancers in detail (Lines 57-68 of the revised manuscript)

  1. I would suggest revising the abstract, make it more connected.

Reply: According to the reviewer’s suggestion, we revised the abstract.

  1. L12-15: Revise this sentence and change Helicobacter pylori to the italic font.    Line 30, 115, 118, 120, 121, 124, 126,  128, 129, 132, Change thought-out the manuscript “Helicobacter pylori to italic font”.

Reply: In the original version (Word file) of our manuscript, “Helicobacter pylori” is shown as the italic font, but incorrectly converted to the normal text during the process of PDF production. We corrected them in the revised manuscript.

  1. Colorectal cancer is the fourth most common cancer and the fifth most common cause of cancer-related death worldwide.

Reply: We corrected the erroneous description about the incidence of colon and rectal cancers in the revised manuscript (Lines 45-46)

  1. L126: Do you mean 5.95? Revise.

Reply: We corrected the erroneous description in the revised manuscript (Lines 135-136)

  1. The author contribution section is not updated.

Reply: We uploaded the author contribution section in the revised version as follows;.

The authors contributed equally to all aspects of the article.

Reviewer 2 Report

In this review article Nakanishi et al discussed about main pathogenetic pathways of inflammation associated cancers in digestive organs, focusing mainly on colitis-associated cancer (CAC), esophageal adenocarcinoma and gastric cancer. Main comments:

1) Considering the wide spectrum of the topic, Authors forgot to speak about hepatocellular carcinoma in HBV/HCV, pancreatic adenocarcinoma in chronic pancreatitis and cancer in celiac disease.

2) The risk of CAC in UC depends on several factors such as entity of inflammation and disease extension. Please comment.

3) H. pylori does not cause only gastric cancer, but even gastric lymphoma.

4) H. pylori should be always italicized.

5) Role of CagA and EPIYA motifs have not been cited on H. pylori related gastric carcinogenesis.

6) Please check NFkB in line 207.

Author Response

We thank the reviewer for his/her thoughtful comments, which have allowed us to improve the paper. We have addressed the reviewer’s criticisms through the following changes to the manuscript.

  1. Considering the wide spectrum of the topic, Authors forgot to speak about hepatocellular carcinoma in HBV/HCV, pancreatic adenocarcinoma in chronic pancreatitis and cancer in celiac disease.

Reply: We totally agree with the reviewer’s comments.  It is obvious that inflammation plays a critical role in the development of hepatobiliary cancers such as hepatocellular carcinomas.  In the current review article, however, we decided to focus on the linkage between inflammation and gastrointestinal cancers in detail. 

  1. The risk of CAC in UC depends on several factors such as entity of inflammation and disease extension. Please comment.

Reply: According to the reviewer’s suggestion, we revised description about the risk factors of colitis-associated colorectal cancers in detail (Lines 57-68 of the revised manuscript)

  1. H.pylori does not cause only gastric cancer, but even gastric lymphoma.

Reply; According to the reviewer’s comments, we added the description about the role of H.pylori infection on the development of gastric lymphoma in the revised manuscript (Lines 128-130)

  1. H.pylori should be always italicized.

Reply: In the original version (Word file) of our manuscript, “Helicobacter pylori” is shown as the italic font, but incorrectly converted to the normal text during the process of PDF production. We corrected them in the revised manuscript.

  1. Role of CagA and EPIYA motifs have not been cited on  pylorirelated gastric carcinogenesis.

Reply: According to the reviewer’s comments, we added the description about the CagA and EPIYA motifs of H.pylori on the development of gastric cancer in the revised manuscript (Lines 124-128)

  1. Please check NFkB in line 207.

Reply: We corrected it in the revised manuscript.

Round 2

Reviewer 1 Report

Thank you for your response. Looks much better now. However, avoid abbreviations in the title "Gastric cancer associated with H. pylori infection". 

Author Response

Looks much better now. However, avoid abbreviations in the title "Gastric cancer associated with H. pylori infection". 

Reply; We corrected the erroneous abbreviation in the title. Thank you very much.

Reviewer 2 Report

The description of EPiYA motifs is insufficient. Authors must describe the different risk according to the different sequences and the difference between Western and Asian strains.

Regarding CAC, please include surveillance protocols as recommended by ECCO guidelines on endoscopy.

Author Response

We thank the reviewer for his/her thoughtful comments.

1. The description of EPiYA motifs is insufficient. Authors must describe the different risk according to the different sequences and the difference between Western and Asian strains.

Reply: According to the reviewer’s suggestion, we added the description about the H.pylori strain in the revised manuscript (Lines 131-140).

2. Regarding CAC, please include surveillance protocols as recommended by ECCO guidelines on endoscopy.

Reply: According to the reviewer’s suggestion, we added the description about the ECCO guidelines on endoscopy in the revised manuscript (Lines 67-72).

Round 3

Reviewer 2 Report

Answers were OK